# Development of Phosphoramidite Reagents for the Synthesis of Base-Labile Oligonucleotides Modified with a Linear Aminoalkyl and Amino-PEG Linker at the 3′-End

**DOI:** 10.3390/molecules27238501

**Published:** 2022-12-03

**Authors:** Takashi Osawa, Qin Ren, Satoshi Obika

**Affiliations:** 1Graduate School of Pharmaceutical Sciences, Osaka University, Yamadaoka 1-6, Osaka 565-0871, Japan; 2Institute for Open and Transdisciplinary Research Initiatives, Osaka University, Yamadaoka 1-3, Osaka 565-0871, Japan; 3National Institutes of Biomedical Innovation, Health and Nutrition, 7-6-8 Saito-Asagi, Osaka 567-0085, Japan

**Keywords:** base-labile oligonucleotide, 3′-modification, amino linker, conjugate

## Abstract

Oligonucleotides with an amino linker at the 3′-end are useful for the preparation of conjugated oligonucleotides. However, chemically modified nucleosides, which are unstable under basic conditions, cannot be incorporated into oligonucleotides using the conventional method entailing the preparation of oligonucleotides bearing a 3′-amino linker. Therefore, we designed Fmoc-protected phosphoramidites for the synthesis of base-labile oligonucleotides modified with a 3′-amino linker. The resultant phosphoramidites were then successfully incorporated into oligonucleotides bearing a 3′-amino linker. Various basic solutions were investigated for protecting group removal. All the protecting groups were removed by treating the oligonucleotides with 40% aqueous methylamine at room temperature for 2 h. Thus, the deprotection time and temperature were significantly reduced compared to the conventional conditions (28% NH_3_ aq., 55 °C, 17 h). In addition, the oligonucleotide protecting groups could be removed using a mild base (e.g., 50 mM potassium carbonate methanol solution). Furthermore, base-labile oligonucleotides bearing an amino linker at the 3′-end were successfully synthesized using the developed phosphoramidite reagents, highlighting the utility of our strategy.

## 1. Introduction

Oligonucleotides conjugated with diverse molecules have attracted substantial research interest in several fields, including oligonucleotide medicine [1,2,3,4], DNA-encoded libraries [5,6], and DNA-based nanotechnology [7,8,9]. Various chemical reactions have been applied for the preparation of conjugated oligonucleotides, including amide bond formation [10,11], thiol Michael addition [12,13,14], copper-catalyzed alkyne-azide cycloaddition (CuAAC) [15,16,17,18], the Diels-Alder reaction [19], Staudinger ligation [20], and oxime ligation [21]. For amide bond formation, conjugated oligonucleotides can generally be obtained using oligonucleotides with linear amino linkers, carboxylic acids, and suitable condensing agents. Structurally diverse oligonucleotides are readily accessible owing to the availability of a large variety of carboxylic acids. In addition, most conjugations are performed at either the 3′- or 5′-end of oligonucleotides. In particular, 3′-modification is advantageous because oligonucleotides with an amino linker at the 3′-end show high nuclease resistance [22].

Oligonucleotides modified with linear 3′-alkylamino linkers are generally synthesized using controlled pore glass (CPG)-immobilized phthalimide-based linkers (Figure 1A) [23,24]. In this method, phthalimide is removed to yield amines after treatment with aqueous ammonia for the deprotection of phosphate and nucleobase moieties. This conventional method can be used to synthesize most of the oligonucleotides with linear 3′-amino linkers. However, as phthalimide ring-opening by aqueous ammonia requires prolonged heating (55 °C, 17 h), this method is not suitable for the synthesis of oligonucleotides with chemically modified nucleosides, which are unstable under basic conditions. As another approach, oligonucleotides bearing linear 3′-aminoalkyl linkers can also be obtained utilizing CPG relying on cleavage by UV irradiation based on *o*-nitrobenzyl chemistry [25]. While this method presumably allows for the introduction of base-labile nucleosides into oligonucleotides, the exposure of oligonucleotides to UV light leads to undesired reactions, such as photocycloaddition of pyrimidines and photo-oxidation of guanine [26,27]. Therefore, we considered that the development of a synthetic method via which both the modification of a base-labile nucleoside and the introduction of an amino linker at the 3′-end can be achieved, would be valuable for accessing oligonucleotide conjugates.

The Fmoc group is a commonly used amine protecting group, especially in peptide synthesis, and it is readily removable by treatment with a weak base, such as 20% piperidine in DMF [28,29,30]. Moreover, phosphoramidites bearing an Fmoc-protected amino group have been used for oligonucleotide synthesis [31,32,33,34,35,36,37,38]. Therefore, we designed Fmoc-based phosphoramidite reagents (Figure 1B) and applied them to oligonucleotide synthesis. By introducing these reagents into oligonucleotides, we considered that the protective group of the terminal amine could be removed under mild basic conditions to provide base-labile oligonucleotides with linear 3′-amino linkers. The amidite reagents were envisaged to be accessed from aminoalcohol **A** and 9-fluorenylmethanol derivative **B**, which can be derived from 4-hydroxymethylbenzaldehyde **C** and fluorene **D**, respectively (Figure 1). Herein, we report the syntheses of phosphoramidites **6a**–**c** and oligonucleotides modified with various 3′-amino linkers as well as base-labile nucleosides.

## 2. Results and Discussion

### 2.1. Synthiesis of Phosphoramidite Reagents

In this study, phosphoramidite reagents **6a**–**c**, which can be applied to incorporate an aminopropyl linker, aminohexyl linker, or amino-PEG2 linker at the 3′-end of oligonucleotides, respectively, were prepared. The syntheses of phosphoramidites **6a**–**c** are summarized in Figure 2. The TBS-protected 4-hydroxymethylbenzaldehyde **1** [39] was treated with the 9-fluorenyl anion generated in situ from fluorene and *n*-butyl lithium, to afford 9-fluorenylmethanol derivative **2** in 84% yield. Using 4-nitrophenyl chloroformate and amino alcohols **3a**–**c**, the obtained alcohol **2** was converted to the corresponding carbamates **4a**–**c**. The primary alcohols in **4a**–**c** were protected by the DMTr group, followed by the removal of the TBS group by treatment with triethylamine trihydrofluoride complex (3HF·Et_3_N) to yield benzyl alcohols **5a**–**c** in a high yield. Finally, phosphoramidites **6a**–**c** were synthesized via phosphitylation of **5a**–**c**.

### 2.2. Optimum Deprotection Conditions for Oligonucleotides Modified with the Developed Phosphoramidite Reagents

Oligonucleotides were synthesized using the designed phosphoramidites and the deprotection efficiency of various base-treatment conditions was investigated. In this experiment, phosphoramidite **6a** and thymidine-immobilized CPG were used and the C10-mer was selected as the model sequence (Figure 3). In addition, because *N*4-benzoylcytidine is known to undergo replacement at the 4-position upon treatment with a base other than aqueous ammonia, such as aqueous methylamine or 50% ethylenediamine in EtOH [40,41], *N*4-acetylcytidine was used, as it is less susceptible to such side reactions [40]. The application of common oligonucleotide synthetic procedures resulted in good stepwise coupling yields (>95%), as estimated based on trityl monitoring. The synthesized oligonucleotide was treated with various basic solutions to remove the protecting groups (Table 1) and the obtained crude oligonucleotide was analyzed using LC/MS (Appendix A).

Initially, the synthesized oligonucleotide was subjected to general deprotection conditions for nucleobase protecting groups, namely treatment with aqueous ammonia at 55 °C for 17 h. This resulted in the complete removal of all protecting groups (Table 1, entry 1). Therefore, the reaction temperature was lowered from 55 °C to room temperature and the reaction time required to remove the protecting group of the 3′-aminopropyl linker was investigated. When the reaction time was 2 h, approximately 20% of the 11-mer oligonucleotide with the Fmoc-protecting group intact remained (Table 1, entry 2 and Appendix A). On the other hand, the protecting group of the aminopropyl linker was completely removed after 8 h (Table 1, entry 3 and Appendix A). To increase the deprotection efficiency, an aqueous solution of methylamine, which is a primary amine generally used for the deprotection of synthesized oligonucleotides [41,42], was used (Table 1, entry 4). The deprotection proceeded at the same temperature and time as described in entry 2. Thus, the reaction time was significantly reduced compared to the conventional conditions (28% NH_3_ aq., 55 °C, 17 h) [23,24].

To enable the application of phosphoramidites **6a**–**c** in the synthesis of oligonucleotides modified with 3′-amino linkers and base-labile nucleosides, deprotection conditions using weak basic solutions relative to aqueous methylamine were investigated (Table 1, entries 5–8). The use of a 50 mM potassium carbonate methanol solution, which is a general mild deprotection reagent for synthetic oligonucleotides, resulted in the complete removal of the protecting groups after 24 h at room temperature (Table 1, entries 5 and 6). All the oligonucleotide protecting groups were removed using 0.4 M sodium hydroxide solution in methanol and water (4:1) after 4 h at room temperature (Table 1, entry 7). Moreover, this condition allowed for the deprotection of nucleobase and phosphate groups, as well as carboxylate hydrolysis [43]. In addition, because thioamides cannot be cleaved under the deprotection conditions described in entry 7, this enables the introduction of thioAmNA, developed by us, into oligonucleotides, as cleavage of the thioamide bridge moiety would be avoided [44]. Moreover, a mixed system comprising *t*-BuNH_2_/MeOH/H_2_O (1:1:2), used for deprotecting TAMRA-T-modified oligonucleotides, which are known to decompose upon treatment with aqueous ammonia [45], was also capable of removing all the oligonucleotide protecting groups (entry 8).

Aminopropyl, hexylamino, and PEG2-amino linkers were incorporated at the 3′-end of 13-mer DNA containing all four nucleobases (A, G, C, and T) using phosphoramidites **6a**–**c** (Figure 4). The treatment of the synthesized oligonucleotides with 40% aqueous methylamine solution (Table 1, entry 4) resulted in the removal of the nucleobase, phosphate, and amino linker protecting groups (Appendix A). In addition, 0.4 M sodium hydroxide solution, which is applicable for the removal of the oligonucleotide amino linker protecting group (Table 1, entry 7), could also be applied for the deprotection of 13-mer DNA containing the three 3′-amino linkers (Appendix A).

### 2.3. Synthesis of Oligonucleotides Modified with 3′-Amino Linkers and Base-Labile Nucleosides

Under the optimized conditions, oligonucleotides modified with 3′-amino linkers and base-labile nucleosides were synthesized. 5-Iodo-2′-deoxyuridine (IdU) [46], 2′,4′-amido-bridged nucleic acid (AmNA) [47,48], and TAMRA-T [45] were chosen as chemically modified nucleosides in these experiments. The nucleosides with a pyrimidine nucleobase substituted with an iodo or bromo group at the 5-position can be converted to 5-aminopyrimidine by heating in aqueous ammonia. The amido-bridge moieties in AmNA undergo cleavage under strongly basic conditions and rhodamine analogs such as TAMRA-T are insufficiently stable under the same. IdU, AmNA, and TAMRA-T were incorporated into the 13-mer DNA modified with an amino linker at the 3′-end. The conditions for oligonucleotide deprotection are summarized in Table 2. The IdU-modified oligonucleotides were treated with 28% aqueous ammonia at room temperature for 8 h to afford the desired oligonucleotides in high yields without the decomposition of 5-iodouracil nucleobase (entries 1–3 in Table 2, Appendix A). After treatment with 50 mM potassium carbonate in methanol, all the protecting groups in the AmNA-modified oligonucleotide were removed without cleavage of the amide bond (entries 4–6 in Table 2, Appendix A). A mixed solution of *t*-BuNH_2_/MeOH/H_2_O (1:1:2), which can be used for the deprotection of TAMRA-T-modified oligonucleotides (entry 8 in Table 1), was applied for the deprotection of oligonucleotides bearing 3′-amino linkers as well as TAMRA-T (entries 7–9 in Table 2, Appendix A). Thus, the use of phosphoramidite reagents **6a**–**c** enabled the isolation of the desired modified oligonucleotides in moderate yields (Table 2).

Furthermore, crude 13-mer DNAs modified with IdU, AmNA, and TAMRA-T synthesized via the conventional method, using commercially available phthalimide-modified CPG, and via our strategy, using phosphoramidite **6a**, were analyzed for comparison employing HPLC. The HPLC and LC/MS plots of the crude DNAs are shown in Figure 2 and Appendix A. In the conventional method, treatment with aqueous ammonia at 55 °C for 17 h for phthalimide removal leads to the formation of side products due to the reactivity of the chemically modified nucleic acid [23,24]. In particular, the yield of the TAMRA-T-modified oligonucleotide was significantly lower than that obtained via the developed method, being <10% based on the peak area in the HPLC chart (Figure 2C). Furthermore, AmNA was completely converted to 4′-carbamoyl-2′-methylaminothymidine via ammonolysis of the amide bridge moiety (Figure 2B). In contrast, as the protecting groups of the 13-mer DNA synthesized using phosphoramidite **6a** were removable with mild base treatment, no conspicuous side reactions were observed. These results demonstrated the utility of the synthetic method developed in this study.

## 3. Materials and Methods

### 3.1. General

All moisture-sensitive reactions were conducted in well-dried glassware under N_2_ atmosphere. The analytical thin layer chromatography (TLC) was applied to monitor reaction progress using pre-coated glass sheets (Silica gel 60 F254, Merck, Darmstadt, Germany). For column chromatography, silica gel PSQ-100B and DIOL MB-40/75 (Fuji Silysia Chemical, Aichi, Japan) were used. The infrared (IR) spectra were recorded on a FT/IR-4200 spectrometer (JASCO, Tokyo, Japan). ^1^H, ^13^C, and ^31^P NMR spectra were recorded using a JNM-ECA500 spectrometer(JEOL, Tokyo, Japan). The chemical shifts were reported in parts per million (ppm) relative to internal tetramethylsilane (δ = 0.00 ppm) and residual CHCl_3_ (δ = 7.26 ppm) for ^1^H NMR. For ^13^C NMR, the chemical shifts were reported in ppm relative to chloroform-*d*_1_ (δ = 77.0 ppm). For ^31^P NMR, the chemical shifts were reported in ppm relative to 5% H_3_PO_4_ (δ = 0.0 ppm) as the external standard. The matrix-assisted laser desorption/ionization–time of flight (MALDI-TOF) mass spectra of all new compounds were recorded on a SpiralTOF JMS-S3000 (JEOL, Tokyo, Japan) instrument. The oligonucleotides were synthesized on a 0.2-μmol scale using an automated DNA synthesizer (nS-8, Gene Design, Osaka, Japan). For analytical HPLC CBM-20A (SHIMADZU, Kyoto, Japan), DGU-20A_3R_ (SHIMADZU, Kyoto, Japan), LC-20AD (SHIMADZU, Kyoto, Japan), CTO-20A (SHIMADZU, Kyoto, Japan), SPD-20A (SHIMADZU, Kyoto, Japan), and SIL-20A (SHIMADZU, Kyoto, Japan) were used. For LC/MS analyses, a Xevo G2-XS QTOF (Waters, Milfold, MA, USA) was utilized.

### 3.2. Synthesis of Phosphoramidite Reagents **6a**–**c**

Compound **2**: Under N_2_ atmosphere, *n*-butyl lithium in *n*-hexane (1.6 M, 10 mL, 16 mmol) was added dropwise to a solution of fluorene (2.66 g, 16 mmol) in dry THF (100 mL) at −78 °C. After stirring for 1 h, compound **1** [39] (4.0 g, 16 mmol) dissolved in dry THF (50 mL) was added dropwise to the solution. The reaction mixture was stirred at −78 °C for 1 h. After quenching with 1% aqueous HCl at −78 °C, the resulting mixture was diluted with EtOAc. The solution was washed with water and brine, dried over Na_2_SO_4_, and concentrated in vacuo. The residue (6.60 g) was purified using column chromatography (silica gel 120 g, CHCl_3_:EtOAc = 5:1) to yield **2** as a white powder (5.60 g, 84%). IR (KBr): *ν*_max_ 3560, 3379, 3065, 3038, 2953, 2927, 2883, 2856, 1513, 1471, 1462, 1447, 1421, 1389, 1374, 1362, 1298, 1255, and 1212 cm^−1^. ^1^H NMR (500 MHz, CDCl_3_) δ: 1.96 (d, *J =* 3.5 Hz 1H), 4.37 (d, *J* = 6.5 Hz, 1H), 4.76 (s, 2.1H), 5.01 (dd, *J* = 3.5, 6.5 Hz, 1H), 7.02–7.04 (m, 1H), 7.14–7.34 (m, 9H), and 7.68–7.71 (m, 2H). ^13^C NMR (125 MHz, CDCl_3_) δ: −5.2, 18.4, 26.0, 54.7, 64.8, 76.2, 119.7, 119.8, 125.6, 125.8, 126.3, 126.6, 126.6, 126.7, 127.5, 127.6, 140.8, 141.1, 141.7, 141.8, 143.3, and 143.7. HRMS (ESI): calcd. for C_27_H_32_NaO_2_Si [MNa^+^] 439.2069, found 439.2064.

Compound **4a**: Under N_2_ atmosphere, pyridine (1.7 mL, 21.2 mmol) and 4-nitrophenyl chloroformate (1.57 g, 6.69 mmol) were added to a solution of **2** (2.95 g, 7.08 mmol) in dry CH_2_Cl_2_ (70 mL) at room temperature. After stirring for 1 h, 3-amino-1-propanol **3a** (1.2 mL, 15.6 mmol) was added to the solution and stirring was continued at room temperature for 48 h. The resulting mixture was diluted with CHCl_3_ and the solution was washed with saturated NaHCO_3_ aq., water, and brine. The collected organic layer was dried over Na_2_SO_4_ and concentrated in vacuo. The residue (5.28 g) was purified using column chromatography (silica gel 100 g, CHCl_3_:EtOAc = 4:1 to 2:1) to yield **4a** as a colorless oil (2.85 g, 78%). IR (KBr): *ν*_max_ 3329, 3064, 3037, 2953, 2883, 2856, 1700, 1611, 1516, 1471, 1448, 1427, 1377, 1361, 1336, and 1256 cm^−1^. ^1^H NMR (500 MHz, CDCl_3_) δ: 0.07 (s, 6H), 0.92 (s, 9H), 1.60–1.63 (m, 2H), 2.63 (s, 1H), 3.22–3.31 (m, 2H), 3.52–3.53 (m, 2H), 4.47 (d, *J* = 6.5 Hz, 1H), 4.70 (s, 2H), 5.09 (t, *J =* 6.5 Hz, 1H), 6.07 (d, *J* = 6.5 Hz, 1H), 7.07–7.39 (m, 10H), and 7.64–7.66 (m, 2H). ^13^C NMR (125 MHz, CDCl_3_) δ: −5.2, 18.4, 25.9, 32.6, 37.4, 52.4, 59.2, 64.7, 77.8, 119.7, 119.8, 123.8, 125.6, 125.8, 125.9, 126.5, 126.7, 127.0, 127.5, 127.6, 136.1, 137.0, 141.1, 141.6, 141.8, 142.5, 143.3, 149.8, and 156.6. HRMS (ESI): calcd. for C_31_H_39_NNaO_4_Si [MNa^+^] 540.2546, found 540.2543.

Compound **4b**: Under N_2_ atmosphere, pyridine (1.4 mL, 17.0 mmol) and 4-nitrophenyl chloroformate (1.26 g, 6.23 mmol) were added to a solution of **2** (2.36 g, 5.66 mmol) in dry CH_2_Cl_2_ (60 mL) at room temperature. After stirring for 1 h, 6-amino-1-hexanol **3b** (1.46 g, 12.5 mmol) was added to the solution and stirring was continued at room temperature for 48 h. The resulting mixture was diluted with CHCl_3_ and the solution was washed with saturated NaHCO_3_ aq., water, and brine. The collected organic layer was dried over Na_2_SO_4_ and concentrated in vacuo. The residue (4.28 g) was purified using column chromatography (silica gel 100 g, CHCl_3_:EtOAc = 4:1 to 2:1) to yield **4b** as a colorless oil (2.15 g, 68%). IR (KBr): *ν*_max_ 3417, 3341, 3066, 2929, 2858, 2738, 1705, 1608, 1520, 1462, 1448, 1404, 1376, 1361, 1336, and 1254 cm^−1^. ^1^H NMR (500 MHz, CDCl_3_) δ: 0.06 (s, 6H), 0.92 (s, 9H), 1.24–1.53 (m, 8H), 3.08–3.13 (m, 2H), 3.54–3.55 (m, 2H), 4.46 (d, *J* = 6.0 Hz, 1H), 4.68 (s, 2H), 4.89 (s, 1H), 6.04 (d, *J =* 6.0 Hz, 1H), and 7.06–7.65 (m, 12H). ^13^C NMR (125 MHz, CDCl_3_) δ: −5.2, 18.4, 25.2, 25.9, 26.2, 29.8, 32.5, 40.7, 52.4, 62.5, 64.7, 77.5, 119.6, 119.7, 125.5, 125.8, 126.0, 126.4, 126.7, 127.1, 127.5, 127.5, 137.1, 141.0, 141.5, 141.8, 142.6, 143.5, and 155.6. HRMS (ESI): calcd. for C_34_H_45_NNaO_4_Si [MNa^+^] 582.3016, found 582.3011.

Compound **4c**: Under N_2_ atmosphere, pyridine (0.35 mL, 4.3 mmol) and 4-nitrophenyl chloroformate (319 mg, 1.58 mmol) were added to a solution of **2** (600 mg, 1.44 mmol) in dry CH_2_Cl_2_ (15 mL) at room temperature. After stirring for 1 h, amino-PEG2 **3c** (473 mg, 3.17 mmol) was added to the solution and stirring was continued at room temperature for 48 h. The resulting mixture was diluted with CHCl_3_ and the solution was washed with saturated NaHCO_3_ aq., water, and brine. The collected organic layer was dried over Na_2_SO_4_ and concentrated in vacuo. The residue (1.18 g) was purified using column chromatography (silica gel 20 g, CHCl_3_:EtOAc = 4:1 to 1:1) to yield **4c** as a colorless oil (540 mg, 63%). IR (KBr): *ν*_max_ 3419, 3328, 3064, 2951, 2928, 2857, 1707, 1533, 1471, 1462, 1449, 1423, 1389, 1376, 1361, 1350, and 1254 cm^−1^. ^1^H NMR (500 MHz, CDCl_3_) δ: 0.06 (s, 6H), 0.92 (s, 9H), 2.37 (t, *J* = 5.5 Hz, 1H), 3.31–3.34 (m, 2H), 3.51–3.73 (m, 10H), 4.48 (d, *J* = 6.5 Hz, 1H), 4.69 (s, 2H), 5.50 (t, *J* = 5.5 Hz, 1H), 6.07 (d, *J =* 6.5 Hz, 1H), 7.09–7.45 (m, 10H), and 7.65–7.66 (m, 2H). ^13^C NMR (125 MHz, CDCl_3_) δ: −5.2, 18.4, 25.9, 40.1, 52.4, 61.8, 64.7, 70.1, 70.4, 72.5, 77.7, 119.6, 119.8, 125.6, 125.8, 126.0, 126.5, 126.7, 127.1, 127.5, 137.1, 141.0, 141.6, 142.6, 143.5, and 155.6. HRMS (ESI): calcd. for C_34_H_45_NNaO_6_Si [MNa^+^] 614.2914, found 614.2908.

Compound **5a**: Under N_2_ atmosphere, 4,4′-dimethoxytrityl chloride (1.98 g, 5.84 mmol) was added to a solution of **4a** (2.75 g, 5.31 mmol) in dry pyridine (50 mL) at room temperature and the reaction mixture was stirred for 12 h. After quenching with MeOH, the resulting mixture was diluted with ethyl acetate. The solution was washed with water and brine, dried over Na_2_SO_4_, and concentrated in vacuo. The residue (6.25 g) was dissolved in dry THF (50 mL) under N_2_ atmosphere, followed by the addition of 3HF·Et_3_N (2.6 mL, 16 mmol) at room temperature and stirring for 24 h. The resulting mixture was diluted with EtOAc and the solution was washed with saturated NaHCO_3_ aq., water, and brine. The collected organic layer was dried over Na_2_SO_4_ and concentrated in vacuo. The residue (5.22 g) was purified using column chromatography (silica gel 100 g, CHCl_3_:EtOAc = 10:1 to 5:1) to yield **5a** as white foam (3.60 g, 96% for 2 steps). IR (KBr): *ν*_max_ 3416, 3061, 3006, 2931, 2874, 2836, 1708, 1607, 1579, 1508, 1447, 1337, 1300, and 1251 cm^−1^. ^1^H NMR (500 MHz, CDCl_3_) δ: 1.71–1.74 (m, 1H), 1.82–1.86 (m, 2H), 3.17–3.26 (m, 4H), 3.72–3.75 (m, 7H), 4.45 (d, *J* = 6.5 Hz, 1H), 4.57 (d, *J* = 5.5 Hz, 2H), 5.27 (t, *J* = 5.5 Hz, 1H), 6.04 (d, *J* = 6.5 Hz, 1H), 6.80–6.82 (m, 4H), and 7.02–7.62 (m, 21H). ^13^C NMR (125 MHz, CDCl_3_) δ: 25.6, 29.3, 39.5, 52.4, 55.2, 61.9, 65.0, 68.0, 77.4, 86.3, 113.2, 115.5, 119.6, 119.8, 125.8, 126.2, 126.3, 126.4, 126.8, 126.8, 127.5, 127.6, 127.9, 128.1, 129.4, 130.0, 136.2, 137.7, 140.3, 141.5, 141.8, 142.5, 143.4, 145.0, 155.4, and 158.4. HRMS (ESI): calcd. for C_46_H_43_NNaO_6_ [MNa^+^] 728.2988, found 728.2986.

Compound **5b**: Under N_2_ atmosphere, 4,4′-dimethoxytrityl chloride (1.36 g, 4.03 mmol) was added to a solution of **4b** (2.05 g, 3.66 mmol) in dry pyridine (30 mL) at room temperature, and the reaction mixture was stirred for 12 h. After quenching with MeOH, the resulting mixture was diluted with ethyl acetate. The solution was washed with water and brine, dried over Na_2_SO_4_, and concentrated in vacuo. The residue (4.18 g) was dissolved in dry THF (30 mL) under N_2_ atmosphere, followed by the addition of 3HF·Et_3_N (1.8 mL, 11 mmol) at room temperature and stirring for 24 h. The resulting mixture was diluted with EtOAc, and the solution was washed with saturated NaHCO_3_ aq., water, and brine. The collected organic layer was dried over Na_2_SO_4_ and concentrated in vacuo. The residue (3.85 g) was purified using column chromatography (silica gel 60 g, CHCl_3_:EtOAc = 10:1 to 5:1) to yield **5b** as yellow foam (2.14 g, 78% over 2 steps). IR (KBr): *ν*_max_ 3403, 3064, 3006, 2933, 2862, 2837, 1706, 1607, 1580, 1509, 1447, 1417, 1337, 1299, and 1250 cm^−1^. ^1^H NMR (500 MHz, CDCl_3_) δ: 1.21–1.44 (m, 5H), 1.57–1.61 (m, 1H), 1.73–1.75 (m, 1H), 1.83–1.86 (m, 1H), 3.00–3.12 (m, 4H), 3.73–3.77 (m, 7H), 4.48 (d, *J* = 6.5 Hz, 1H), 4.58 (d, *J* = 6.0 Hz, 2H), 4.78–4.79 (m, 1H), 6.12 (d, *J =* 6.5 Hz, 1H), 6.80–6.82 (m, 4H), 7.08–7.44 (m, 19H), and 7.61–7.64 (m, 2H). ^13^C NMR (125 MHz, CDCl_3_) δ: 25.6, 26.0, 26.6, 29.8, 30.0, 41.0, 52.3, 55.2, 63.3, 65.0, 68.0, 85.6, 112.9, 119.7, 119.8, 125.7, 126.0, 126.3, 126.5, 126.6, 126.7, 127.3, 127.6, 127.7, 128.2, 130.0, 136.7, 137.7, 140.3, 141.5, 141.8, 142.6, 143.3, 145.4, 155.5, and 158.2. HRMS (ESI): calcd. for C_49_H_49_NNaO_6_ [MNa^+^] 770.3458, found 770.3454.

Compound **5c**: Under N_2_ atmosphere, 4,4′-dimethoxytrityl chloride (327 mg, 0.97 mmol) was added to a solution of **4c** (520 mg, 0.88 mmol) in dry pyridine (10 mL) at room temperature and the reaction mixture was stirred for 12 h. After quenching with MeOH, the resulting mixture was diluted with ethyl acetate. The solution was washed with water and brine, dried over Na_2_SO_4_, and concentrated in vacuo. The residue (1.05 g) was dissolved in dry THF (10 mL) under a N_2_ atmosphere, followed by the addition of 3HF·Et_3_N (0.42 mL, 2.6 mmol) and stirring at room temperature for 24 h. The resulting mixture was diluted with EtOAc and the solution was washed with saturated NaHCO_3_ aq., water, and brine. The collected organic layer was dried over Na_2_SO_4_ and concentrated in vacuo. The residue (928 mg) was purified using column chromatography (silica gel 20 g, CHCl_3_:EtOAc = 4:1 to 2:1) to yield **5c** as white foam (600 g, 88% for 2 steps). IR (KBr): *ν*_max_ 3427, 3370, 3061, 3034, 3006, 2934, 2975, 2837, 1720, 1607, 1580, 1508, 1448, 1420, 1351, 1300, and 1252 cm^−1^. ^1^H NMR (500 MHz, CDCl_3_) δ: 1.62 (t, *J* = 6.5 Hz, 1H), 3.25–3.27 (m, 2H), 3.33–3.35 (m, 2H), 3.53–3.69 (m, 8H), 3.75 (s, 6H), 4.45 (d, *J* = 6.5 Hz, 1H), 4.59 (d, *J* = 6.0 Hz, 2H), 5.37 (t, *J* = 5.5 Hz, 1H), 6.05 (d, *J =* 6.5 Hz, 1H), 6.79–6.81 (m, 4H), 7.06–7.35 (m, 16H), 7.46–7.50 (m, 3H), and 7.62–7.63 (m, 2H). ^13^C NMR (125 MHz, CDCl_3_) δ: 55.2, 63.1, 65.1, 70.1, 70.7, 70.8, 77.7, 86.0, 113.0, 119.6, 119.8, 125.7, 126.3, 126.7, 126.8, 127.5, 127.6, 127.8, 128.2, 130.1, 136.3, 140.3, 142.5, 143.3, 145.0, and 158.4. HRMS (ESI): calcd. for C_49_H_49_NNaO_8_ [MNa^+^] 802.3356, found 802.3368.

Compound **6a**: Under an Ar atmosphere, *N*,*N*-diisopropylethylamine (0.5 mL, 3.0 mmol) and *i*-Pr_2_NP(Cl)O(CH_2_)_2_CN (0.33 mL, 1.5 mmol) were added to a solution of compound **5a** (706 mg, 1.0 mmol) in anhydrous CH_2_Cl_2_ (10 mL) at 0 °C. The reaction mixture was then stirred at room temperature for 1 h. After quenching with saturated NaHCO_3_ aq., the resulting mixture was extracted with CHCl_3_. The combined organic layers were washed with water and brine, dried over Na_2_SO_4_, and concentrated in vacuo. The residue (1.10 g) was purified using column chromatography (silica gel 40 g, *n*-hexane: EtOAc = 3:1) to yield compound **6a** as a white foam (790 mg, 87%). ^1^H NMR (500 MHz, CDCl_3_) δ: 1.14–1.17 (m, 12H), 1.72–1.74 (m, 2H), 2.57–2.59 (m, 2H), 3.17–3.26 (m, 4H), 3.60–3.63 (m, 2H), 3.80–3.82 (m, 8H), 4.43–4.45 (m, 1H), 4.55–4.59 (m, 1H), 4.65–4.70 (m, 1H), 5.21–5.23 (m, 1H), 6.02–6.03 (m, 1H), 6.81–6.84 (m, 4H), 7.01–7.46 (m, 19H), and 7.58–7.64 (m, 2H). ^31^P NMR (202 MHz, CDCl_3_) δ: 149.1 and 149.1. HRMS (ESI): calcd. for C_55_H_60_N_3_NaO_7_P [MNa^+^] 928.4067, found 928.4044.

Compound **6b**: Under Ar atmosphere, *N*,*N*-diisopropylethylamine (0.51 mL, 3.0 mmol) and *i*-Pr_2_NP(Cl)O(CH_2_)_2_CN (0.33 mL, 1.5 mmol) were added to a solution of compound **5b** (748 mg, 1.0 mmol) in anhydrous CH_2_Cl_2_ (10 mL) at 0 °C. The reaction mixture was then stirred at room temperature for 1 h. After quenching with saturated NaHCO_3_ aq., the resulting mixture was extracted with CHCl_3_. The combined organic layers were washed with water and brine, dried over Na_2_SO_4_, and concentrated in vacuo. The residue (1.05 g) was purified using column chromatography (silica gel 40 g, *n*-hexane: EtOAc = 3:1) to yield compound **6b** as a white foam (780 mg, 82%). ^1^H NMR (500 MHz, CDCl_3_) δ: 1.13–1.25 (m, 14H), 1.23–1.43 (m, 4H), 1.58–1.61 (m, 1H), 1.84–1.87 (m, 1H), 2.57–2.59 (m, 2H), 3.01–3.22 (m, 4H), 3.61–3.63 (m, 2H), 3.73–3.82 (m, 8H), 4.47–4.49 (m, 1H), 4.57–4.59 (m, 1H), 4.66–4.68 (m, 1H), 4.76–4.77 (m, 1H), 6.10–6.11 (m, 1H), 6.81–6.83 (m, 4H), 7.08–7.44 (m, 19H), and 7.62–7.65 (m, 2H). ^31^P NMR (202 MHz, CDCl_3_) δ: 149.1 and 149.1. HRMS (ESI): calcd. for C_58_H_66_N_3_NaO_7_P [MNa^+^] 970.4536, found 970.4525.

Compound **6c**: Under Ar atmosphere, *N*,*N*-diisopropylethylamine (0.51 mL, 3.0 mmol) and *i*-Pr_2_NP(Cl)O(CH_2_)_2_CN (0.33 mL, 1.5 mmol) were added to a solution of compound **5c** (780 mg, 1.0 mmol) in anhydrous CH_2_Cl_2_ (10 mL) at 0 °C. The reaction mixture was then stirred at room temperature for 1 h. After quenching with saturated NaHCO_3_ aq., the resulting mixture was extracted with CHCl_3_. The combined organic layers were washed with water and brine, dried over Na_2_SO_4_, and concentrated in vacuo. The residue (988 mg) was purified using column chromatography (silica gel 40 g, *n*-hexane: EtOAc = 2:1) to yield compound **6c** as a white foam (600 mg, 61%). ^1^H NMR (500 MHz, CDCl_3_) δ: 1.13–1.20 (m, 12H), 2.56–2.59 (m, 2H), 3.25–3.84 (m, 22H), 4.44–4.70 (m, 3H), 5.32–5.35 (m, 1H), 6.04–6.05 (m, 1H), 6.80–6.81 (m, 4H), 7.06–7.48 (m, 19H), and 7.62–7.63 (m, 2H). ^31^P NMR (202 MHz, CDCl_3_) δ: 149.1 and 149.1. HRMS (ESI): calcd. for C_58_H_66_N_3_NaO_9_P [MNa^+^] 1002.4434, found 1002.4429.

### 3.3. Synthesis of Oligonucleotides

Phosphoramidites **6a**–**c**, dT-phosphoramidite (Sigma), Ac-dC-phosphoramidite (Sigma), Pac-dA-phosphoramidite (Gren Research), iPrPac-dG-phosphoramidite (Gren Research), IdU-phosphoramidite (Gren Research), and AmNA-phosphoramidite [47] were dissolved in anhydrous MeCN into a final concentration of 0.1 M. TAMRA-T-phosphoramidite (Gren Research) was dissolved in anhydrous THF/MeCN (1:9) into a final concentration of 0.1 M. Oligonucleotide synthesis was performed on a 0.2-μmol scale using an automated DNA synthesizer (Gene Design nS-8) and 0.25 M 5-(ethylthio)-1*H*-tetrazole in MeCN (Sigma) as an activator, 3% trichloroacetic acid in CH_2_Cl_2_ (Wako) as a deblocking reagent, 0.02 M iodine in THF/pyridine/H_2_O (Sigma) as an oxidizing reagent, and a combination of 5% phenoxyacetic anhydride (Pac_2_O) in THF (Sigma) and 16% 1-methylimidazole in THF (Sigma) as a capping reagent. The oligonucleotides synthesized in trityl-on mode were cleaved from the CPG resin and all the protecting groups were removed by treatment with basic solutions, as shown in Table 1 and Table 2. The oligonucleotides were purified using Sep-Pak^®^ Plus C18 cartridges (Waters) and the 5′-DMTr group was removed during purification using 1% (*v*/*v*) aqueous trifluoroacetic acid (TFA). The separated crude oligonucleotides were analyzed using reversed-phase HPLC (Waters XBridge^®^ MS C18 Column 2.5 μm, 4.6 × 50 mm) and a mixture of 0.1 M TEAA aq. as eluent A and MeCN as eluent B. A linear gradient from 5% to 30% MeCN (over 25 min) was applied at a flow rate of 1 mL/min and a temperature of 50 °C and the process was monitored employing UV visualization at 260 nm. The oligonucleotides were further analyzed using reversed-phase LC/MS (Waters BEH C18 Column 1.7 μm, 2.1 × 100 mm) and a mixture of H_2_O/HFIP/Et_3_N (100/1/0.1) as eluent A and MeOH as eluent B. A linear gradient from 5% to 40% MeOH (over 10 min) was used at 50 °C at a flow rate of 0.3 mL/min and the process was monitored employing UV visualization at 260 nm. The separated oligonucleotides were further purified using reversed-phase HPLC (Waters XBridge^®^ OST C18 column 2.5 μm, 10 × 50 mm) and 0.1 M TEAA buffer (pH = 7.0) as eluent A and MeCN as eluent B. A linear gradient from 5% to 30% MeCN (over 25 min) was used at 50 °C at a flow rate of 3 mL/min, and the process was monitored employing UV visualization at 260 nm. The oligonucleotide yield was calculated from the peak values recorded at 260 nm using a NanoDrop instrument (DeNovix DS-11).

## 4. Conclusions

In this study, Fmoc group-based phosphoramidite reagents **6a**–**c** were developed to enable the incorporation of various amino linkers at the 3′-end of oligonucleotides. Phosphoramidite reagents **6a**–**c** were synthesized from aminoalcohols **3a**–**c** and 9-fluorenylmethanol derivative **2**, which was prepared from fluorene and TBS-protected 4-hydroxymethylbenzaldehyde **1**. The reagents were successfully introduced into oligonucleotides and the amino linker protecting group was removed via mild base treatment. The significantly milder reaction conditions compared to those of the conventional method enabled the introduction of base-labile nucleosides as well as various amino linkers at the 3′ end, demonstrating the utility of our method.

## Data Availability

Not applicable.

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
