# Peer review of "Development of Phosphoramidite Reagents for the Synthesis of Base-Labile Oligonucleotides Modified with a Linear Aminoalkyl and Amino-PEG Linker at the 3′-End"

_molecules, 2022, doi:10.3390/molecules27238501_

Round 1
Reviewer 1 Report
This manuscript presented the development of phosphoramidite reagents and strategy for the synthesis of base-labile oligonucleotides. Overall, the data is sound, esp, the stectra of copmounds are clean and the logic is clear, which meet the requirement from molecules. Here, the authors are suggested to detail the significance and reasons to develop a new strategy for synthesis of base-labile oligonucleotides, which would benefit broader readers. Also, please compare this mehod/strategy to the reported ones and comment the pro and con. After addressing these, this manuscript should be published.
Author Response
Point 1: This manuscript presented the development of phosphoramidite reagents and strategy for the synthesis of base-labile oligonucleotides. Overall, the data is sound, esp, the stectra of copmounds are clean and the logic is clear, which meet the requirement from molecules.
Response 1: We are grateful to reviewer 1 for the useful suggestions that have helped us to improve our paper. As indicated in the responses, we have taken all the comments and suggestions into account in the revised version of our paper. Changes made in response to the comments have been marked up using the Track Changes function.
Point 2: Here, the authors are suggested to detail the significance and reasons to develop a new strategy for synthesis of base-labile oligonucleotides, which would benefit broader readers. Also, please compare this mehod/strategy to the reported ones and comment the pro and con. After addressing these, this manuscript should be published.
Response 2: We thank the referee’s comment. According to the comment, we added a comparison of our synthetic method with existing synthetic methods and the importance of developing our phosphoramidites to the introduction. We have been added the sentences as follows.
Revised manuscript (line 47 of page 2 – line 49 of page 2):
This conventional method can be used to synthesize most of oligonucleotides with linear 3′-amino linkers.
Revised manuscript (line 67 of page 2 – line 69 of page 2):
By introducing these reagents into oligonucleotides, we considered that the protective group of the terminal amine could be removed under mild basic conditions to give base-labile oligonucleotides with linear 3′-amino linkers.
In addition, the sentences have been changed as follows.
Revised manuscript (line 54 of page 2 – line 57 of page 2):
As another approach, oligonucleotides bearing linear 3′-aminoalkyl linkers can also be obtained utilizing CPG relying on cleavage by UV irradiation based on o-nitrobenzyl chemistry [25]. While this method presumably allows introduction of base-labile nucleosides into oligonucleotides, exposure of oligonucleotides to UV light leads to undesired reactions, such as photocycloaddition of pyrimidines and photooxidation of guanine [26,27].
Original (line 50 of page 2 – line 54 of page 2):
Although oligonucleotides bearing linear 3′-aminoalkyl linkers can also be obtained utilizing CPG relying on cleavage by UV irradiation based on o-nitrobenzyl chemistry [25], exposure of oligonucleotides to UV light leads to undesired reactions, such as photocycloaddition of pyrimidines and photooxidation of guanine [26,27].
Reviewer 2 Report
The paper describe new reagent and, to some extent, methodology for the synthesis of oligonucleotide conjugates. Novel protecting group proposed in this paper serves to solve the relevant problem of the instability of some oligonucleotides in the cleavage conditions suitable for common PG. The work was conducted on a sufficient scientific level, and all the conclusions are supported by the provided experimental data. Logic of the study is clear for the reader. Citation is consistent and appropriate. This paper is the case when I have no corrections or concerns. I hope the PG described in this paper will meet its application on oligonucleotide chemistry.
Author Response
Point 1: This is an interesting study that presents new Fmoc-protected phosphoramidite monomers for use in the synthesis of base-labile oligonucleotides that are modified with a 3′-amino linker. A number of base-labile oligonucleotides bearing a 3'-aminolinker were synthesized using these phosphoramidite monomers. Importantly the monomers can be used in standard solid-phase oligonucleotide synthesis cycles, making them generally applicable and easy to use. The approach will be of value in the oligonucleotide field.
Response 1: Thank you for taking precious time to review our paper. We are grateful to reviewer 2 for the comments.
Reviewer 3 Report
This is an interesting study that presents new Fmoc-protected phosphoramidite monomers for use in the synthesis of base-labile oligonucleotides that are modified with a 3′-amino linker. A number of base-labile oligonucleotides bearing a 3'-aminolinker were synthesized using these phosphoramidite monomers. Importantly the monomers can be used in standard solid-phase oligonucleotide synthesis cycles, making them generally applicable and easy to use.The approach will be of value in the oligonucleotide field.
Author Response
Point 1: The paper describe new reagent and, to some extent, methodology for the synthesis of oligonucleotide conjugates. Novel protecting group proposed in this paper serves to solve the relevant problem of the instability of some oligonucleotides in the cleavage conditions suitable for common PG. The work was conducted on a sufficient scientific level, and all the conclusions are supported by the provided experimental data. Logic of the study is clear for the reader. Citation is consistent and appropriate. This paper is the case when I have no corrections or concerns. I hope the PG described in this paper will meet its application on oligonucleotide chemistry.
Response 1: Thank you for taking precious time to review our paper. We are grateful to reviewer 3 for the comments.